# In Silico Molecular Analysis of Carbapenemase-Negative Carbapenem-Resistant *Pseudomonas aeruginosa* Strains in Greece

**DOI:** 10.3390/microorganisms12040805

**Published:** 2024-04-16

**Authors:** Katerina Tsilipounidaki, Christos-Georgios Gkountinoudis, Zoi Florou, George C. Fthenakis, Efthymia Petinaki

**Affiliations:** 1Faculty of Medicine, University of Thessaly, 41500 Larissa, Greece; tsilipoukat@gmail.com (K.T.); gountinoudis@gmail.com (C.-G.G.); zoi_fl@yahoo.gr (Z.F.); 2Veterinary Faculty, University of Thessaly, 43100 Karditsa, Greece

**Keywords:** antibiotic resistance, carbapenem, Greece, MexR, MexT, NalC, NalD, OprD porin, *Pseudomonas aeruginosa*

## Abstract

To date, three carbapenem resistance mechanisms have been identified: carbapenemase released from the pathogen, changes in the expression of the outer membrane OprD porin, and overexpression of the efflux pump MexAB-OprM. Twelve carbapenemase-negative carbapenem-resistant *Pseudomonas aeruginosa* strains, isolated from patients hospitalized at the University Hospital of Larissa, Central Greece, during 2023, which belonged to various sequence types (STs), were selected and were studied focusing on the characterization of their *β*-lactamases, on changes to OprD and its regulator MexT proteins, and on alterations to the MexAB-OprM regulator proteins encoded by the *mexR*, *nalC*, and *nalD* genes. Whole genome sequencing analysis revealed the presence of *β*-lactamase encoding genes, with *bla*_PAO_ present in all isolates. Additionally, seven different genes of the oxacillinase family (*bla*_OXA-35_, *bla*_OXA-50_, *bla*_OXA-395_, *bla*_OXA-396_, *bla*_OXA-486_, *bla*_OXA-488_, *bla*_OXA-494_) were identified, with each strain harboring one to three of these. Regarding the OprD, five strains had truncated structures, at Loop 2, Loop 3, Loop 4, and Loop 9, while the remaining strains carried previously reported amino acid changes. Further, an additional strain had a truncated MexR; whereas, two other strains had totally modified NalC sequences. The active form of MexT, responsible for the downregulation of OprD production, as the intact sequence of the NalD protein, was found in all the strains studied. It is concluded that the truncated OprD, MexR, and NalC proteins, detected in eight strains, probably led to inactive proteins, contributing to carbapenem resistance. However, four strains carried known modifications in OprD, MexR, and NalC, as previously reported in both susceptible and resistant strains, a finding that indicates the complexity of carbapenem resistance in *P. aeruginosa*.

## 1. Introduction

The dissemination of carbapenem-resistant *Pseudomonas aeruginosa* (CRPA) is a real threat worldwide [1]. In 2017, the World Health Organization established that CRPA should be considered a critical pathogen of first priority due to its increasing incidence of isolation [2].

In *P. aeruginosa*, resistance to carbapenem is associated mainly with the production of carbapenemases (*bla*_VIM_, *bla*_IMP_, *bla*_KPC_, *bla*_NDM_, *bla*_GES_, etc.) but also with the loss or the decreased production of the outer membrane barrier pore protein (OprD) or the upregulation of the efflux pump (MexAB-OprM) [3]. Although the majority of CRPA strains are carbapenemase producers, carbapenemase-negative isolates have also been identified [4].

OprD is the only pore protein found in *P. aeruginosa* that is conducive to the passage of antibiotics [5]. Structurally, the OprD protein consists of 443 amino acids, which form 16*β*-antiparallel sheets, connected by seven loops on the periplasmic side of the cell outer membrane and eight loops (L1–L8) of variable size located outside of the cell [5]. The chemical characteristics of the amino acids confer the three-dimensional structure of the protein, forming a channel through which antibiotics, such as imipenem, enter into the bacterium. The basic amino acids located in L2 (75G-98S) and L3 (113Q-137A) play a role in imipenem passage. Frequently, the downregulation of *oprD* can happen due to the *mexT* gene, which downregulates OprD at the transcriptional and post-transcriptional levels [6]. Additionally, mutational changes of the *oprD* gene result in the loss of an active porin [6]. Therefore, the downregulation or the loss of OprD usually contributes to resistance to common carbapenem antibiotics (mainly to imipenem). Hence, deletion, insertion, or mutation of the gene encoding OprD can reduce imipenem uptake by *P. aeruginosa*.

The operon MexAB-OprM was the first multidrug efflux pump reported in *P. aeruginosa* and is considered to be the main contributor to antibiotic resistance [7]. MexAB-OprM can export several antibiotics, including quinolones, macrolides, tetracyclines, lincomycin, and most β-lactams [7]. Overexpression of this efflux pump MexAB-OprM was often shown in CRPA, which could be contributing to their carbapenem resistance [7]. MexAB-OprM is constitutively expressed in wild-type strains and its expression is negatively controlled by the repressor genes *mexR*, *nalC*, and *nalD* [8,9,10]. More specifically, transcriptional repression of the mexAB-oprM operon is mediated directly by MexR and NalD proteins and indirectly by NalC, which represses ArmR protein, an anti-repressor of MexR (Appendix A) [11]. Any type of *mexR*, *nalC*, and *nalD* mutants might lead to the up-regulation of MexAB-OprM [12].

In Central Greece, according to recent epidemiological data, 70% of CRPA strains carry either the *bla*_VIM_ or the *bla*_NDM_ gene; whereas, the remaining 30% seem to have a different mechanism of resistance [13]. The objective of the present study was the investigation of the mechanism of resistance of carbapenemase-negative carbapenem-resistant *P. aeruginosa* strains, focusing on the characterization of their β-lactamases combined with the changes in proteins encoded by the *oprD*, *mexT*, *mexR*, *nalC*, and *nalD* genes.

## 2. Materials and Methods

### 2.1. Collection of Isolates

Twelve strains of *P. aeruginosa*, all isolated from patients hospitalized at the University Hospital of Larissa, which is located in Central Greece, during 2023, were included in this study based on (a) resistance profiles to at least one carbapenem (imipenem or meropenem); (b) RAPIDEC^®^ CARBA NP negative results; (c) the absence of *bla*_VIM_, *bla*_KPC_, *bla*_NDM_, and *bla*_OXA_-48 genes; and (d) a variety of sequence types (STs 664, 446, 235, 299, 162, 253, 2048, 110, 4312). In addition, one *P. aeruginosa* strain fully susceptible to carbapenems (Strain 4) and two *P. aeruginosa* strains that are carbapenemase positive and carbapenem resistant (Strains 22, 23) were also included and studied as control strains for comparison.

### 2.2. Susceptibility Testing and Molecular Characterization

The bacteria were identified and subjected to susceptibility testing using the automated Vitek II system (bioMérieux, Marcy l’ Etoile, France). Minimal inhibitory concentration (MIC) values of imipenem, meropenem, and ceftazidime–avibactam were re-determined by using the MIC test strip (Lofilchem, Roseto degli Abruzzi, Italy); the MIC of colistin was determined by the broth microdilution method (ComASP™, Lofilchem), following the EUCAST guidelines (https://www.eucast.org, accessed on 1 February 2024). The RAPIDEC^®^ Carba-NP test (bioMérieux) was performed in all isolates according to instructions of the manufacturer [14]. The presence of carbapenemase-encoding genes, including *bla*_VIM_, *bla*_NDM_, *bla*_KPC_, and *bla*_OXA-48_, was assessed as previously described [15]. Molecular typing of the isolates was conducted using MultiLocus Sequence Typing (MLST), involving the amplification of seven gene loci (*acsA*, *aroE*, *guaA*, *mutL*, *nuoD*, *ppsA*, *trpE*) by PCR, as per established protocols (https://pubmlst.org/organisms/pseudomonas-aeruginosa/primers, accessed on 1 February 2024). Subsequently, all isolates underwent further characterization through Whole Genome Sequencing (WGS).

### 2.3. Whole Genome Sequence

The libraries for genomic DNA were prepared using Ion Torrent Technology and Ion Chef Workflows (Thermo Fisher Scientific, Waltham, MA, USA). Subsequently, sequencing of the genomic DNA libraries was performed on the S5XLS system, followed by primary data analysis using Ion Torrent Suite (v.5.10.0). Quality assessment of the reads was conducted using FastQC software (v.0.11.9); assembly of the reads was performed using the SPAdes genome assembler (v3.15.5) with the default settings. The assembled genomes’ quality was evaluated using the Quast version 5.2.0 tool and average coverage was determined for each genome using the mapPacBio tool from BBTools (https://sourceforge.net/projects/bbmap/; accessed on 9 January 2024). The nucleotide alterations of the genes *oprD*, *mexR*, *nalC*, and *nalD* were confirmed by PCR, followed by sequencing analysis, using primers designed for the purpose of this study.

For the identification of genes associated with antibiotic resistance, ResFinder-4.4.2 was employed with the ID threshold set to 90% and the minimum length set to 60%. Comparative analysis of the genome between strains was performed using Blast analysis, using as reference the *P. aeruginosa* PAO1 genome (accession no AE004091.2).

### 2.4. Genomic Analysis of Carbapenem Resistance Mechanisms

The respective genes involved in carbapenem resistance (*oprD*, *mexT*, *mexR*, *nalC*, *nalD)* were characterized using publicly available programs. Initially, the sequence of each gene was translated into amino acids utilizing the Expasy tool (https://web.expasy.org/translate/, accessed on 1 February 2024). Following translation, each resulting protein was compared to the *P. aeruginosa* genome PAO1 using the ClustalW (https://www.genome.jp/tools-bin/clustalw, accessed on 1 February 2024) tool, in order to identify any amino acid changes, using the default parameters.

### 2.5. Data Management and Analysis

Data were entered into Microsoft Excel and analyzed using SPSS v. 21 (IBM Analytics, Armonk, NY, USA). A basic descriptive analysis was performed. Potential associations between the presence of carbapenem resistance in a strain, the MIC for imipenem or meropenem, and the number of β-lactamase encoding genes detected in a strain were evaluated by using the Pearson chi-square test or Fisher exact test, as appropriate, and by Spearman’s rank correlation.

### 2.6. Nucleotide Accession Numbers

The genomes of *P. aeruginosa* strains have been deposited in GenBank under BioProject accession PRJNA1084717.

## 3. Results

### 3.1. Results of Susceptibility Testing

The antimicrobial profiles of the strains studied are in Table 1. Nine strains showed resistance to both imipenem (MIC > 4 μg/mL) and meropenem (MIC > 8 μg/mL), two strains showed resistance to imipenem only, and one strain showed resistance to meropenem only. All the strains were susceptible to ceftazidime–avibactam and to colistin whilst five strains were found to be resistant to aztreonam (MIC > 16 μg/mL).

### 3.2. Presence of β-Lactamase Encoding Genes

Whole genome sequencing analysis revealed the presence of β-lactamase encoding genes, with *bla*_PAO_ present in all isolates. Additionally, seven different genes of the oxacillinase family (*bla*_OXA-35_, *bla*_OXA-50_, *bla*_OXA-395_, *bla*_OXA-396_, *bla*_OXA-486_, *bla*_OXA-488_, *bla*_OXA-494_) were identified, with each strain harboring one to three of these. Of these β-lactamases, none have a confirmed activity as carbapenemases.

### 3.3. Amino Acid Alterations of the OprD Porin and MexT Protein

Five strains, which belonged to different sequence types (ST162, ST235, ST664, ST2048), had several alterations (mutations, deletions, or insertions), which resulted in truncated proteins. Specifically, two strains (nos. 20 and 21) shared a common OprD structure comprising 93 identical amino acids with some changes, such as S59T (DRVDWT_61–66_ TASTGP) and A93V in respective strains. Another two strains (nos. 3 and 6) contained an identical structure of 141 amino acids, which, in Strain 6, was continued with a chain of 77 amino acids, similar to that of strains with accession numbers BBA54061 and LC321998. Strain 8 had a deletion of 20 amino acids. The comparison of these structures with respective ones in Strain PAO1 is in Figure 1.

The remaining strains had several alterations, which also led to amino acid changes; all these had been previously reported [16]. Specifically, four patterns have been found: Pattern 1 (V127L), Pattern 2 (T103S, K115T, F170L, E185Q, P186G, V189T, R310E, A315G, G425A), Pattern 3 (D43N, S57E, S59R, E202Q, I210A, E230K, S240T, N262T, A267S, A281G, K296Q, Q301E, R310G, V359L, _372_-VDSSSSYAGL-_383_), and Pattern 4 (S57E, S59R, V127L, E185Q, P186G, V189T, E202Q, I210A, E230K, S240T, N262T, T276A, A281G, K296Q, Q301E, R310E, A315G, L347M, _372_-VDSSSSYAGL-_383_, S403A, Q424E) (Table 2). It is noted that Pattern 1 was previously reported in a strain recovered in India (ETD93478.1); Pattern 2 was reported in strains recovered in India (MH135308.1) and the United States of America (CP000438.1); Pattern 3 was reported in strains recovered in Spain (MH050333), the United States of America (KY086497.1), Lebanon (KJ482586.1), Iran (KT239372.1), and Canada (CP007224.1); and Pattern 4 was reported in strains in India (MH 135309.1, MH 135311.1), as found in the NCBI database.

Among the control strains, the carbapenem-susceptible strain (no. 4) had an intact OprD; the two carbapenem-resistant strains (nos. 22 and 23) also had alterations, all of which had been described above.

Analysis of the nucleotide sequences of the mexT-encoding gene showed the presence of a deletion of 8-bp (CGGCCAGC) compared to that of PAO1, which was present in all 12 strains. This deletion is responsible for the conversion of the inactive MexT to the active form, as previously described [17]. The active MexT can reduce the production of OprD. The three control strains also had the deletion described above.

### 3.4. Amino Acid Alterations of MexR, NalC, and NalD Proteins

The MexR protein was identical to that of PAO1 in four strains; another six strains showed known amino acid changes (Table 2). One isolate had alterations, which led to a truncated protein of 84 amino acids, of which the initial 23 were identical to those of PAO1; another strain had a protein structure of 145 amino acids, of which the initial 88 amino acids were identical to that of PAO1 (Figure 2).

With regard to the NalC protein, one strain had a truncated protein of 42 amino acids and another one a protein of 214 amino acids; in both cases, the amino acid sequences were totally different from that of PAO1 (Figure 2, Appendix A). Probably, these altered proteins lead to the de-repression of the ArmR, which binds to *mexR* and changes its composition, releasing the mexA-mexB-oprM efflux pump expression. The remaining strains had known alterations. Finally, all isolates had a NalD structure identical to that of PAO1.

Among the control strains, the susceptible strain (no. 4) had intact the NalC and NalD, similar to those in PAO1. Among the resistant strains, the most important finding was a shortening of NalD (Strain 22), which comprised 80 amino acids.

### 3.5. MICs and Carbapenems with Truncated OprD, MexR, and NalC Proteins

Among the nine strains fully resistant to both carbapenems, five had a truncated OprD structure whilst among the three strains that were not fully resistant to both antibiotics, no such findings were seen (*p* = 0.08). Among these three strains, which were resistant either to imipenem or to meropenem, one had a truncated NalC and a second one had a NalC totally different from that of PAO1 (Table 2). No other associations or correlations were found during statistical analysis (*p* > 0.20 for all other comparisons).

## 4. Discussion

The mechanism of carbapenem resistance among carbapenemase-negative CRPA strains has attracted the interest of researchers as it is evidently complicated and multifactional. In such strains, β-lactamase overproduction, OprD alterations, and overexpression of efflux pumps can co-exist [18,19].

In the present study, the first step was the characterization of β-lactamase, given that some oxacillinases can act as carbapenemases (OXA-23-like, OXA-24/40-like, OXA-48-like, OXA-58-like, OXA-143-like, and OXA-235) [20,21]. However, no such genes have been detected in the strains evaluated in the present study.

In addition, as previously reported, the decrease or loss of the expression of OprD porin, due to polymorphisms, insertion sequences, or deletions, can affect the entry of imipenem and thus contribute to carbapenem resistance [22]. Most of the amino acid substitutions (shown in Table 2) have been previously reported [23,24,25]. However, their role in carbapenem resistance has not been fully elucidated and, thus, they have been considered to be ‘’irrelevant modifications’’ [25]. These modifications could have a potential role given that a recent study has indicated that the polymorphism F170L or a specific shortening in Loop 7, both detected in carbapenem-susceptible strains, were associated with the development of resistance to carbapenem in the future under continuous selection pressure conditions [26].

Among the twelve strains studied in this work, five fully resistant had truncated OprDs. Two of them had premature stop codons at Loop 2, one at Loop 3, and one at Loop 4; whereas, the fifth one had a 20-amino-acid deletion at the terminal of Loop 8. Although Loops 2 and 3 are essential for antibiotic entry, the shortening of the other loops might also affect the porin channel, leading to a decreased penetration [27]. Similar results have also been published in a study of *P. aeruginosa* strain isolates in Korea [28]. These findings indicate that, potentially, the low permeability of drugs due to the mutational inactivation of OprD could be responsible for carbapenem resistance.

MexR, the product of the regulator gene *mexR*, is a transcriptional repressor of the mexA-mexB-oprM efflux pump operon [29]. Mutations in *mexR* leading to amino acid substitution at the 8th, 37th, 44th, 54th, 55th, 66th, 126th, and 138th positions have been reported by many authors [30,31]. Choudhury et al. [29] have reported the formation of a stop codon at the 35th position, which resulted in the termination of the polypeptide and could lead to overexpression of the MexAB-OprM efflux pump. Among the strains studied in this work, one strain, which was fully resistant to both imipenem and meropenem, had mutations that caused the premature termination of the peptide at the 84th position and led to the formation of an altered protein. Hence, we hypothesize that this defective MexR protein cannot act as a repressor because of the loss of its ability.

With regard to the NalC, among the strains studied in this work, only two had mutations that led to 41 and 214 amino acid peptides, respectively, without any similarity with the NalC of PAO1. Probably, these altered structures were unable to downregulate the mexA-mexB-oprM efflux pump operon, contributing to resistance to carbapenem. The most frequent substitution found among the remaining 10 strains, was the G71E, in accord with previous reports [25,32]. The other amino acid substitutions found (A145V, E153Q, A186T, S209R) have also been reported [30,33,34]. However, it has been shown that G71E, A145V, E153Q, A186T, and S209R, detected in NalC, have been described in strains not displaying mexA-mexB-oprM overproduction [23].

Although none of the 12 strains studied in this work carried genes that could encode β-lactamases with carbapenemase activity, we cannot exclude the possibility that they overproduced enzymes, which hydrolyzed carbapenems [4]. Additionally, we also consider that the truncated OprDs and MexRs, such as the altered NalC peptides, might also play a role in carbapenem resistance [31]. The simultaneous susceptibility of these strains to ceftazidime/avibactam is hopeful, given that the metallo-β-lactamase producers’ CRPA strains cannot be treated by using the combination of these antibiotics [35].

## 5. Conclusions

In the present study, we provide some findings regarding the mechanism of resistance to carbapenem among 12 carbapenemase-negative carbapenem-resistance *P. aeruginosa* by means of in silico WGS analysis. During this study, no unusual carbapenemases have been detected; although, we report on some relevant modifications (premature stop codons or frameshifts) in OprD, MexR, and NalC. These alterations might modify their functionality, contributing to resistance to carbapenem.

## Figures and Tables

**Figure 1 microorganisms-12-00805-f001:**
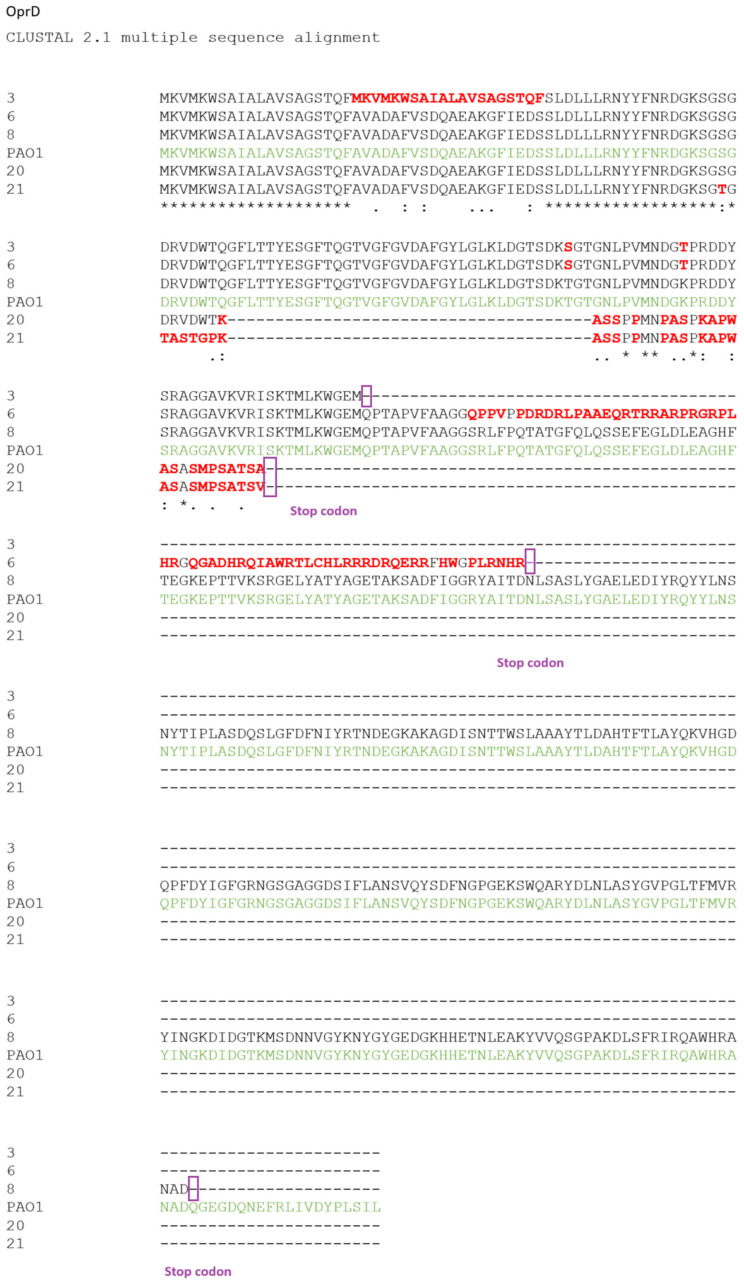
Results of the comparison of the sequence of four *P. aeruginosa* strains compared to the reference sequence of the strain *P. aeruginosa* PAO1, as obtained by using the ClustalW tool. The image depicts the shows strains that exhibit stop codons in the amino acid sequence of OprD (highlighted in purple). Differences between the sequences are indicated in red. * indicates similarity, the others (“:” or “.”) indicate differences between sequences. The green color corresponds to PAO1 amino acid sequence.

**Figure 2 microorganisms-12-00805-f002:**
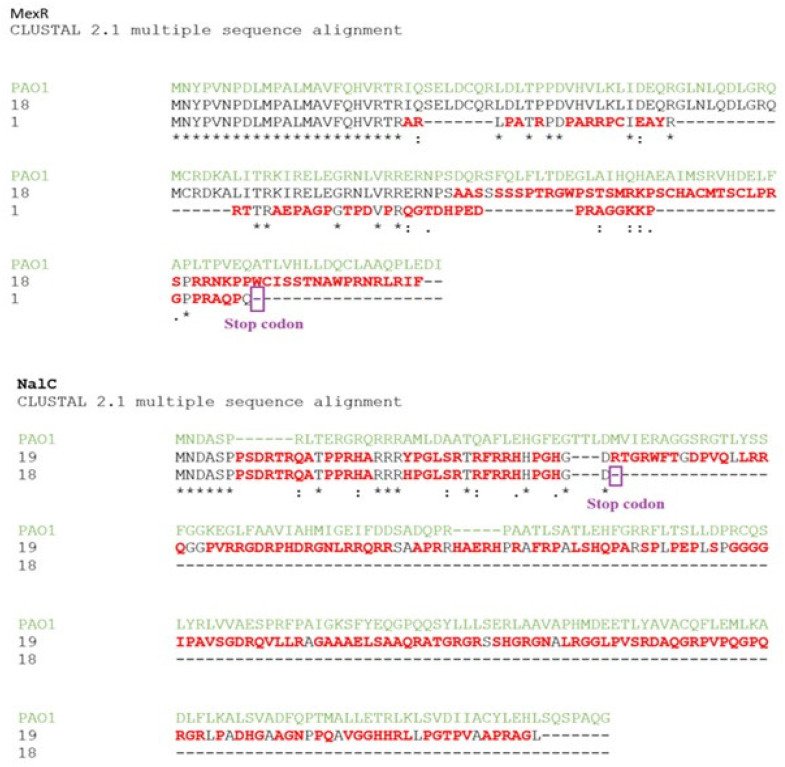
Representation of altered MexR and NalC protein sequences in *P. aeruginosa* strains compared to the PAO1 reference strain, according to the ClustalW tool (top: MexR proteins from one isolate exhibiting truncations resulting in an 83-amino-acid protein while another isolate shows a 145-amino-acid structure; bottom: NalC proteins from the two isolates displaying truncations, with one being 42 amino acids long and the other 214 amino acids long; differences between the sequences indicated in red). ***** indicates similarity. Both “:” and “.” indicate the differences. The green color corresponds to amino-acid sequence of PAO1.

**Table 1 microorganisms-12-00805-t001:** Details of the recovery of 12 carbapenem-resistant and carbapenemase-negative *P. aeruginosa* strains isolated from a University hospital in Central Greece and of their antimicrobial profiles.

Id	ST ^2^	Source	Medical Department of Hospital	Minimum Inhibitory Concentrations (μg/mL) ^1^													
				IMP	MEM	CAZ/AVI	AMK	ATM	FEP	CAZ	CIP	COL	LVX	PIP	PIP/TAZ	TIC/CLA	TOB
1	664	Purulent material	Nephrology	>32	>32	1	4	8	4	8	≤0.25	≤0.5	1	64	32	32	2
2	446	Wound	ICU ^3^	>32	>32	1	16	≥64	≥64	16	1	≤0.5	1	≥128	≥128	≥128	≥16
3	235	Tissue	Orthopaedics	>32	>32	0.5	≥64	16	≥64	4	≥4	≤0.5	≥8	≥128	≥128	≥128	≥16
5	299	Bronchial secretions	ICU	>32	8	0.38	≤2	2	≤1	2	≤0.25	≤0.5	0.5	≤4	≤4	16	2
6	162	Sputum	Respiratory Medicine	>32	>32	1	≥64	16	8	32	1	≤0.5	4	≥128	≥128	>128	≥16
7	253	Bronchial secretions	ICU	>32	>32	1.5	4	≥64	8	8	0.5	≤0.5	4	64	32	>128	2
8	2048	Tissue	ICU	>32	>32	1	≤2	16	8	4	≤0.25	≤0.5	1	32	32	>128	2
17	110	Bronchial secretions	ICU	16	16	0.75	≤2	4	4	4	≤0.25	1	0,5	8	≤4	32	2
18	253	Wound	Oncology	3	16	3	4	≥64	16	16	≥4	1	≥8	64	64	≥128	2
19	4312	Bronchial secretions	Respiratory Medicine	16	8	2	≤2	16	4	4	≤0.25	≥16	1	16	8	64	2
20	664	Venous catheter	Internal Medicine	12	>32	2	≥64	32	16	4	≥4	0.5	≥8	16	16	≥128	≥16
21	235	Wound	Neuro-surgery	>32	>32	2	≥64	≥64	≥64	4	≥4	1	≥8	≥128	≥128	≥128	≥16
4	244	Pleural fluid	ICU	0.38	1	0.75	≤2	16	2	4	0.5	≤0.5	2	16	8	32	≤1
23	308	Bronchial secretions	Internal Medicine	>32	>32	0.38	≥64	≥64	≥64	≥64	≥4	1	≥8	≥128	≥128	>128	≥16
24	395	Blood	ICU	>32	>32	0.38	≥64	16	16	≥64	≥4	≤0.5	≥8	64	64	>128	≥16

^1^ IMP: imipenem, MEM: meropenem, CAZ/AVI: ceftazidime/avibactam, AMK: amikacin, AZM: aztreonam, CPM: cefepime, CEF: ceftazidime, CIP: ciprofloxacin, COL: colistin, LVX: levofloxacin, PIP: piperacillin, PIP/TAZ: piperacillin/tazobactam, TIC/CLA: ticarcillin/clavulanic acid, TOB: tobramycin; ^2^ sequence type (in MultiLocus Sequence Typing); ^3^ intensive care unit.

**Table 2 microorganisms-12-00805-t002:** Presentation of β-lactamase genes and alterations in OprD, NalC, NalD, and MexR protein sequences present among 12 carbapenem-resistant carbapenemase-negative *P. aeruginosa* strains.

			Test Strains												Control Strains		
Isolate No.			1	2	3	5	6	7	8	17	18	19	20	21	4	22	23
Sequence type ^1^			664	446	235	299	162	253	2048	110	253	4312	664	235	244	308	395
Minimum inhibitory concentrations (μg/mL)		IMP ^2^	>32	>32	>32	>32	>32	>32	>32	16	3	16	12	>32	0.38	>32	>32
		MEM ^2^	>32	>32	>32	8	>32	>32	>32	16	16	8	>32	>32	1	>32	>32
β-Lactamase encoding genes			*bla*_PAO_, *bla*_OXA-50_	*bla*_PAO_, *bla*_OXA-395_	*bla*_PAO_, *bla*_OXA-35_, *bla*_OXA-488_	*bla*_PAO_, *bla*_OXA-50_, *bla*_OXA-396_, *bla*_OXA-494_	*bla*_PAO_, *bla*_OXA-50_, *bla*_OXA-396_, *bla*_OXA-494_	*bla*_PAO_, *bla*_OXA-488_	*bla*_PAO_, *bla*_OXA-396_, *bla*_OXA-494_	*bla*_PAO_, *bla*_OXA-486_	*bla*_PAO_, *bla*_OXA-488_	*bla*_PAO_, *bla*_OXA-396_, *bla*_OXA-494_	*bla*_PAO_, *bla*_OXA-50_	*bla*_PAO_, *bla*_OXA-35_, *bla*_OXA-488_	*bla*_PAO_, *bla*_OXA-396_, *bla*_OXA-494_	*bla*_PAO_, *bla*_OXA-10_, *bla*_OXA-488_, *bla*_NDM-1_	*bla*_PAO_, *bla*_OXA-10_, *bla*_OXA-488_, *bla*_VIM-2_
Alterations ^3^	OprD	D43N			STOP CODON		STOP CODON		STOP CODON	+			STOP CODON	STOP CODON	WT		+
		S57E								+		+					+
		S59R								+		+					+
		T103S		+		+		+			+					+	
		K115T		+		+		+			+					+	
		V127L	+									+					
		F170L		+		+		+			+					+	
		E185Q		+		+		+			+	+				+	
		P186G		+		+		+			+	+				+	
		V189T		+		+		+			+	+				+	
		E202Q								+		+					+
		I210A								+		+					+
		E230K								+		+					+
		S240T								+							+
		N262T								+		+					+
		A267S								+							+
		T276A										+					
		A281G								+		+					+
		K296Q								+		+					+
		Q301E								+		+					+
		R310E		+		+		+		+	+					+	+
		A315G		+		+		+			+	+				+	
		L347M										+					
		V359L								+							+
		372VDSSS- SYAGL383								+		+					+
		S403A										+					
		Q424E										+					
		G425A		+		+		+			+					+	
	NalC	G71E	+	+	+	+	+	+	+	+	STOP CODON	CHANGE	+	+	WT	+	+
		D79E														+	
		A145V		+				+									
		E153Q			+									+			
		A186T				+				+							
		S209R	+	+	+		+	+	+				+	+		+	
	NalD		WT	WT	WT	WT	WT	WT	WT	WT	WT	WT	WT	WT	WT	DELE TION	WT
	MexR	G101E	STOP CODON			WT	WT	WT	+	WT	CHANGE						WT
		V126E		+	+				+			+	+	+	+	+	

^1^ In MultiLocus Sequence Typing; ^2^ IMP: imipenem, MEM: meropenem; ^3^ Alterations compared to the reference sequence of *P. aeruginosa* PAO1. + indicates positivity.

## Data Availability

All the data associated with this manuscript are provided within the manuscript.

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
