# Peer review of "In Silico Molecular Analysis of Carbapenemase-Negative Carbapenem-Resistant Pseudomonas aeruginosa Strains in Greece"

_microorganisms, 2024, doi:10.3390/microorganisms12040805_

Round 1

Reviewer 1 Report

Comments and Suggestions for Authors

The main question addressed by the research: molecular analysis of antibiotic resistant, especially carbapenem-resistant Pseudomonas aeruginosa strains.

This topic «In silico molecular analysis of carbapenem-resistant carbapenemase-negative Pseudomonas aeruginosa strains in Greece» relevant in the field of the molecular genetic characteristics of this microorganism, its antibiotic resistance and combating antibiotic Resistant infection.

This article summarized information about molecular mechanisms of carbapenem-resistant Pseudomonas aeruginosa strains.

The advantage of this article - the genomes of P. aeruginosa strains have been deposited in GenBank.

The conclusion helps the reader evidence and arguments presented for understand the important point of molecular mechanisms of carbapenem-resistant P. aeruginosa strains in Greece.

The references appropriate. The number of references (34) is enough, in addition, the number of sources five years ago (2019-2024) is 44,1% (15), which is enough.

The figures are informative and illustrative.

Disadvantage of the article:

1.      In this article very small number of strains - twelve strains of P. aeruginosa.

2.      In the international nomenclature of bacteria, the abbreviated name should be written correctly - P. aeruginosa.

3.      No information about statistical methodology and no statistical evidence of the data.

4.      There is no conclusion of the ethical commission on the possibility of conducting this study.

Author Response

Reviewer 1:

Thank you very much for your useful comments.

  1. In this article very small number of strains - twelve strains of P. aeruginosa.

As our initial aim was to see what happen with the carbapenem resistance in these strains we selected a small number of strains, hoping that, after the elaboration of the results, the study will be continued with more specific experiments focused on the expression of proteins involved.

  1. 2.   In the international nomenclature of bacteria, the abbreviated name should be written correctly - P. aeruginosa.

The corrections have been done.

  1. No information about statistical methodology and no statistical evidence of the data.

Some statistical analysis was performed in an attempt to find associations within this dataset. However, there only a tendency of association, which has bene described in the revised manuscript.

  1. There is no conclusion of the ethical commission on the possibility of conducting this study.

      The study refers to bacterial strains recovered during routine laboratory investigation of samples sent to our Department. Patients were not involved in this study and no clinical information are presented or demographic characteristics are disclosed. Hence, in accord with relevant Greek and European Union legislation, no specific licence is needed for the publication of these findings.

Reviewer 2 Report

Comments and Suggestions for Authors

In the manuscript „In silico molecular analysis of carbapenem-resistant carbapenemase-negative Pseudomonas aeruginosa strains in Greece” the Authors describe the genome analysis of 12 antibiotic-resistant Pseudomonas aeruginosa isolates. The antibiotic resistance of P. aeruginosa is a sensitive issue in healthcare, and the presented results can be interesting to the scientific community. However, I found some issues concerring the way the results are presented, making it harder to understand them. I would suggest the Authors to modify the manuscript and make it clearer.

Major suggestions:

1. Abstract should be made clearer, and should present the main agenda with more details. It remains unclear in the Abstract why oprD and other genes were analysed – I would suggest to shortly mention their possible function in resistance. Also, it is unclear what is meant here by “irrelevant modifications”.

2. In both Abstract and main text, the choice of the isolates is presented as based several factors, one of them “sequence type” (lines 16, 77). The term is too broad. The Authors should clearly state that it was MLST types they are referring to. 

3. Statement in lines 40-41 should be supported by reference.

4. In the introduction, OprD is presented by describing its structure in the text. It would be clearer, if protein structure, or predicted model should be presented as a figure; in this case the mutations detected could be also indicated more clearly on the protein.

5. Line 39 it is unclear what is meant by “so-called nfxC mutants”.

6. I would encourage the Authors to also present the protein interaction/regulation scheme, in order for the reader to understand the connection among the analysed genes better.

7. Statement in lines 58-60 should be supported by reference.

8. Not enough information is given about the control isolates – it is unclear which is which in Table 2; also there is no MIC information concerning theses strains (and PAO1 also).

9. Unclear when and why two different methods for MIC detection were used. Please indicate more precisely in the text.

10. In line 105 the Authors present that gaps were confirmed by PCR and sequencing – it would be useful to know how many of the gaps were remaining, and are there any indication as to why.

11. The ID threshold in ResFinder was set to 90% and the minimum length set to 60%; what was the reason of these exact numbers?

12. Line 115 why is PAO1 called international genome? Perhaps it was meant reference genome?

13. In chapter 3.3 versions of the proteins are called “structures”, I would suggest using terms like truncated protein, or some other variant, as the term “structures” does not reveal what was observed clearly.

14. Why were protein sequences compared, but not nucleic acid sequences? It should be presented clearly in the text.

15. What could be the reason of control isolates, the ones that have carpapenemases, to have alterations in genes analysed in this article?

16. In lines 196-197 the Authors claim that the susceptible control strain had “had intact the MexR, NalC and NalD, similar to those in PAO1”. However, in Table 2, this control strain has mutation indicated in MexR (as mentioned above (comment 8), it is unclear in this table, which strain is actually susceptible – I think it is 4?).

17. In the Conclusions, I would argue that the observations made by the Authors could not be considered “evidence” yet (line 261), as only targeted mutagenesis could prove the phenotype is associated with the genotype. I would suggest re-phrasing.

Minor revisions:

1. There are some grammar/typing errors in the manuscript. The Authors should look through carefully. Some errors that I have noticed: lines 25 “may probably” is redundant; line 26 I would suggest “protein inactivation”; line 38 extra comma; line 53 unclear meaning “result to loss”; line 64 I would suggest to use “mutations”; line 262 should be “negative”.

2. Inconsistent writing of beta (e.g. line 60), and gene names (should be italic) line 252.

Comments on the Quality of English Language

There are some grammar/typing errors in the manuscript. The Authors should look through carefully. Some errors that I have noticed: lines 25 “may probably” is redundant; line 26 I would suggest “protein inactivation”; line 38 extra comma; line 53 unclear meaning “result to loss”; line 64 I would suggest to use “mutations”; line 262 should be “negative”.

Author Response

Reviewer 2:

Thank you very much for your useful comments.

  1. Abstract should be made clearer, and should present the main agenda with more details. It remains unclear in the Abstract why oprD and other genes were analysed – I would suggest to shortly mention their possible function in resistance. Also, it is unclear what is meant here by “irrelevant modifications”.

The abstract was modified according to your suggestions.

  1. In both Abstract and main text, the choice of the isolates is presented as based several factors, one of them “sequence type” (lines 16, 77). The term is too broad. The Authors should clearly state that it was MLST types they are referring to. 

We added the STs.

  1. Statement in lines 40-41 should be supported by reference.

The reference was added “Hammami S, Ghozzi R, Burghoffer B, Arlet G, Redjeb S. Mechanisms of carbapenem resistance in non-metallo-beta-lactamase-producing clinical isolates of Pseudomonas aeruginosa from a Tunisian hospital. Pathol. Biol. 2009; 57:530–535”

  1. In the introduction, OprD is presented by describing its structure in the text. It would be clearer, if protein structure, or predicted model should be presented as a figure; in this case the mutations detected could be also indicated more clearly on the protein.

We need a permission and it is difficult to obtain it at this stage of the manuscript. However, the structure of OprD is referred in the reference 6.

  1. Line 39 it is unclear what is meant by “so-called nfxC mutants”.

The sentence was changed.

  1. I would encourage the Authors to also present the protein interaction/regulation scheme, in order for the reader to understand the connection among the analyzed genes better.

A scheme regarding the regulation of mexA-mexB-oprM by the respective genes was added as supplementary figure.

  1. Statement in lines 58-60 should be supported by reference.

The reference was added.

  1. Not enough information is given about the control isolates – it is unclear which is which in Table 2; also there is no MIC information concerning theses strains (and PAO1 also).

All these information regarding the tree control strains have been added in the text. Unfortunately, the reference strain PAO1 has been only in silico analysed.

  1. Unclear when and why two different methods for MIC detection were used. Please indicate more precisely in the text.

Given that some of these strains expressed resistance only to imipenem or meropenem and in order to ensure correctly their MICs levels we have used two methods (automated and Etest)

  1. 10. In line 105 the Authors present that gaps were confirmed by PCR and sequencing – it would be useful to know how many of the gaps were remaining, and are there any indication as to why.

We rephrased  the sentence such as: The nucleotide alterations of genes oprD, mexR, nalC and nalD were confirmed by PCR followed by sequencing analysis, using primers designed for the purpose of this study.

  1. The ID threshold in ResFinder was set to 90% and the minimum length set to 60%; what was the reason of these exact numbers?

These parameters have been used for a reliable analysis.

  1. Line 115 why is PAO1 called international genome? Perhaps it was meant reference genome?

We have changed the international genome by the reference genome.

  1. In chapter 3.3 versions of the proteins are called “structures”, I would suggest using terms like truncated protein, or some other variant, as the term “structures” does not reveal what was observed clearly.

The corrections have been done

  1. Why were protein sequences compared, but not nucleic acid sequences? It should be presented clearly in the text.

We have added the nucleic acid sequences as supplementary material.

  1. What could be the reason of control isolates, the ones that have carbapenemases, to have alterations in genes analysed in this article?

We have included three control strains in order to see if they carry alterations of OprD, MexR, NalC and NalD.

  1. In lines 196-197 the Authors claim that the susceptible control strain had “had intact the MexR, NalC and NalD, similar to those in PAO1”. However, in Table 2, this control strain has mutation indicated in MexR (as mentioned above (comment 8), it is unclear in this table, which strain is actually susceptible – I think it is 4?).

Yes, the control strain has alteration in MexR. The correction has been done.

  1. In the Conclusions, I would argue that the observations made by the Authors could not be considered “evidence” yet (line 261), as only targeted mutagenesis could prove the phenotype is associated with the genotype. I would suggest re-phrasing.

We agree with you and it was rephrased.

Minor revisions:

  1. There are some grammar/typing errors in the manuscript. The Authors should look through carefully. Some errors that I have noticed: lines 25 “may probably” is redundant; line 26 I would suggest “protein inactivation”; line 38 extra comma; line 53 unclear meaning “result to loss”; line 64 I would suggest to use “mutations”; line 262 should be “negative”.
  2. Inconsistent writing of beta (e.g. line 60), and gene names (should be italic) line 252.

All your suggestions have been done.

Reviewer 3 Report

Comments and Suggestions for Authors

This is an interesting study on alternative explanations for carbapenem resistance in Pseudomonas aeruginosa. Even if there are several such studies published earlier the results should be a good starting point for further investigation into the role of the alterations found. That also brings out the weaknesses in the study. First, without genetic experiments it is always hard to link an alteration found with a specific genotype. However, that would require new experimental setup and would be good for a follow up paper if possible. The other issue, which should be easier is to perform RNA-seq to determine alterations in expression of the important proteins that may have influence on the resistance level.

Otherwise, I find the manuscript well written and easy to follow. Just a few minor comments:

line 78-79; "two Ps. aeruginosa strains carbapenemase-positive carbapenem-resistant" could be rephrased as "two carbapenemase-positive and carbapenem-resistant Ps. aeruginosa strains"

line 167; "convention" should probably be "conversion" 

Figures 1 and 2; premature stop codons are noted, and frame shift mutations are inferred by the abrupt change in identity to reference sequence. It would improve the interpretation to get the nucleotide sequence comparisons for the frame shift mutations and some information on the quality control of the reads at the frame shift mutation sites.

Lines 196-198 does not completely match table 2 as I read it. First, the susceptible strain seems to have a point mutation in mexR and the deletion in NalD seems to be in only one of the resistant strains. Please clarify this.

Line 235; a stop point at the end should be removed

Line 238; "which had resulted" could be "which resulted"

Author Response

Reviewer 3:

Thank you very much for your useful comments.

line 78-79; "two Ps. aeruginosa strains carbapenemase-positive carbapenem-resistant" could be rephrased as "two carbapenemase-positive and carbapenem-resistant P. aeruginosa strains"

The correction has been done

line 167; "convention" should probably be "conversion" 

The correction has been done

Figures 1 and 2; premature stop codons are noted, and frame shift mutations are inferred by the abrupt change in identity to reference sequence. It would improve the interpretation to get the nucleotide sequence comparisons for the frame shift mutations and some information on the quality control of the reads at the frame shift mutation sites.

We have added the nucleic acid sequences as supplementary material. The quality control of the reads at the frame shift mutation sites was verified by PCR combined with sequence analysis.

Lines 196-198 does not completely match table 2 as I read it. First, the susceptible strain seems to have a point mutation in mexR and the deletion in NalD seems to be in only one of the resistant strains. Please clarify this.

Yes, deletion of NalD was observed in the resistant strain. The control strain has alteration in MexR and the correction has been done.

Line 235; a stop point at the end should be removed

It was corrected

Line 238; "which had resulted" could be "which resulted"

It was corrected